# Progress and Stagnation of Renovation, Energy Efficiency, and Gentrification of Pre-War Walk-Up Apartment Buildings in Amsterdam Since 1995

**Leo Oorschot and Wessel De Jonge \***

Heritage & Design, Section Heritage & Architecture, Department of Architectural Engineering & Technology, Faculty of Architecture and the Built Environment, Delft University of Technology, Julianalaan 134, 2628BL Delft, The Netherlands; l.m.oorschot@tudelft.nl
\* Correspondence: w.dejonge@tudelft.nl

**Abstract:** Increasing the energy efficiency of the housing stock has been one of the largest challenges of the built environment in the Netherlands in recent decades. Parallel with the energy transition there is an ongoing revaluation of the architectural quality of pre-war residential buildings. In the past, urban renewal was traditionally based on demolition and replacement with new buildings. This has changed to the improvement of old buildings through renovation. Housing corporations developed an approach for the deep renovation of their housing stock in the period 1995–2015. The motivation to renovate buildings varied, but the joint pattern that emerged was quality improvement of housing in cities, focusing particularly on energy efficiency, according to project data files from the NRP institute (Platform voor Transformatie en Renovatie). However, since 2015 the data from the federation of Amsterdam-based housing associations AFWC (Amsterdamse Federatie Woningcorporaties) has shown the transformation of pre-war walk-up apartment buildings has stagnated. The sales of units are slowing down, except in pre-war neighbourhoods. Housing associations have sold their affordable housing stock of pre-war property in Amsterdam inside the city's ring road. The sales revenue was used to build new affordable housing far beyond the ring road. This study highlights the profound influence of increasing requirements established by the European Energy Performance of Building Directive (EPBD) and the revised Housing Act of 1 July 2015, for the renovation of the pre-war housing stock. The transformation process to climate-neutral neighbourhoods inside the ring road is slowing down because of new property owners, making a collective heat network difficult to realize; furthermore, segregation of residents is appearing in Amsterdam.

**Keywords:** walk-up apartment building; pre-war; Amsterdam; energy efficiency; gentrification; segregation

---

## 1. Introduction

### 1.1. The Challenge

Increasing the energy efficiency of the housing stock has been one of the largest challenges in the built environment of the Netherlands and many other countries in recent decades. In line with the international Paris-Climate-Change-Conference 2015, Dutch cities have great ambitions to increase energy savings and reduce greenhouse gas emissions by 2050. According to the European Energy Performance of Building Directive (EPBD) [1], all public buildings and all new buildings must meet the Nearly Zero-Energy Buildings (NZEBs) norm with very high energy performance by 2019 and 2020, respectively. The Dutch translation for NZEBs is BENG [2]. The principle of nearly zero-energy buildings means a building that has very high energy performance. To determine this, the NZEB or BENG uses three indicators to calculate the energy quality of buildings. The maximum energy

consumption in kWh/m$^2$ usable floor space a year is one. The maximum primary fossil energy use in kWh/m$^2$ usable floor space a year is the second indicator, and thirdly, the minimal fraction of renewable energy as a per centage.

Vereniging van woningcorporaties Aedes, the Dutch Association of Social Housing Organizations, has promised that their average affordable housing stock will have Energy Performance Certificate (EPC) label B by 2021 and label A by 2030 [3]. The recently published housing agenda 2017–2021 of Aedes promises a climate-neutral housing stock for all Dutch housing associations by 2050 [4].

The newly appointed government recently presented a climate agreement as policy for the upcoming years with a reduction of carbon emissions for the built environment of 3.4 megaton by 2030 and carbon neutrality by 2050 [5].

In the period 1995–2015 the reduction of carbon emission in relation to geopolitics was already an ambition [6], but the life cycle of buildings or building components and the carbon footprint were not considered. At this time there was only a focus on the energy efficiency of buildings [7]. This challenge resulted in all kinds of renovation programs of the existing housing stock of housing associations in the Netherlands and consequently, living expenses have increased for tenants.

### 1.2. Object of Research

The object of this research is the tenement four-storey apartment buildings of housing associations dating to before the Second World War. They are called walk-up apartment buildings because they do not have lifts. Once they were the flagships of the early Dutch welfare state and they provided many Dutch people with an affordable place to live in cities. The buildings are usually part of a perimeter city block of three to four-storeys and the façade is directly on the building line. Buildings follow carefully designed urban spaces. For this reason, urban blocks can be folded or irregular, leading to exceptions in the floorplans of corner apartments. The gardens inside the blocks are usually private, although some are known to be collective. The balconies of the units are shallow. All storys, including the plinth, are made up of apartments [8–10]. In Amsterdam there are usually four stacked units located on a closed stairwell that share one service shaft. The four-storey volumes have an urban character, typically featuring a pitched roof over an attic with storage space. In Den Haag there are usually six units located around an open stairwell and two service shafts. The blocks have a flat roof, are usually three storys high and are less urban in their architectural character and size. A somewhat similar type was built in Rotterdam, where the bedrooms were always located on the top floor under a flat roof. This type had three or four storys per stairwell and the apartments were usually deep because each one had its own internal staircase to the bedroom floor. In Den Haag and Rotterdam the front door was supposed to be located directly on the street or at least on open-air. The units were usually small. In the inter-war period, about 35 per cent of the apartments had one bedroom, 31–34 per cent had two bedrooms, 21–25 per cent had three bedrooms, and 6 per cent had four bedrooms. These units are nowadays valuable real estate objects and attractive places to live in the urban areas.

### 1.3. Questions

In this study the object of research is the pre-war walk-up apartment buildings inside the ring road of Amsterdam and the phenomena of stagnation of deep renovation of this housing stock. Therefore, the main research questions are: What was the successful approach to renovation in the period 1995–2015? What is the influence of the European regulations Energy Performance of Building Directive (EPBD) on the renovation stagnation in Amsterdam? How is the renovation stagnation influenced by national legislation (such as the change of the Housing Act of 2015)? What is the cause of the increase in living expenses, such as rent and energy costs? What are the consequences for housing associations from these developments? [11]. What are the consequences for Amsterdam and other municipalities from these developments?

*1.4. Method*

The method that was applied in this research is a survey of annual statistics provided by the federation of housing associations of Amsterdam, Amsterdamse Federatie Woningcoporaties (AFWC), about their housing stock and tenants, and annual project data files provided by the institute NRP (Platform voor Transformatie en Renovatie) about deep renovation projects. Furthermore, documents published by the government, housing associations, municipalities, consultancies, and institutes were used as sources to interpret the data. These documents clarify why and in which way these buildings were renovated, and why changes took place.

NRP is a Dutch institute and organization that deals with knowledge and education about the renovation process of buildings and the transformation of the built environment. It is supported by real estate companies, the Delft University of Technology, architects, consultancies, housing associations, and construction companies. Each year, the NRP organizes the Gulden Feniks Award and collects project data files about the renovation of this housing stock from the participants competing for the award [7].

Another important source is the annual statistics about the housing stock and tenants, which are provided by the AFWC. This is the federation of the nine Amsterdam based housing associations. In 2018 they reportedly owned 186,429 dwellings units in that city [12]. This is about 42% of the total amount of dwellings in Amsterdam. These statistics present key information about migration, segregation, and diversification, and social housing in Amsterdam. The data is used by the municipality, tenants organization of HA (Huurdersvereniging Amsterdam), and housing associations to reach an agreement about the local housing policy in accordance with national legislation.

In addition, we paid attention to the renewed Housing Act of 2015, which had an impact on the housing market in the Netherlands. Primary literature was studied to understand the changes in the practice of renovation of the housing stock, for example protocols about the renovation strategies of housing associations, such as housing association Eigen Haard.

This study brings different fields of knowledge (statistics, documents, protocols, project data files) together and draws conclusions about the stagnation of the renovation of pre-war apartment buildings of housing associations.

*1.5. Terms*

There is no European standard or definition of terms dealing with changes to buildings. Usually these terms are defined by each member country. Renovation monitoring on a European scale is, for that reason, difficult. However, terms used in relation to the change of residential buildings are: maintenance, refurbishment, renovation, retrofitting, preservation, and restoration. According to the Cambridge Dictionary, the definitions of these terms (with their Dutch equivalents) are as follows.

Maintenance (onderhoud)—the work needed to keep a road, building, machine, etc., in good condition. Refurbishment (opknappen)—work such as cleaning, repairing, painting, and decorating to make a building look new again. Renovation—the act or process of repairing and improving something, especially the quality of a building. Retrofitting (technisch aanpassen)—the act of providing a machine with a part, or a place with equipment, which the machine or place did not have when it was first built. Preservation (behoud)—the act of keeping something the same or of preventing it from being damaged. Restoration (restauratie naar een oorspronkelijke staat)—the act or process of returning something to its earlier good condition or position.

Looking from the perspective of Dutch housing associations these terms are defined in several protocols provided by housing associations and used by architects and consultancies [8–10]. Maintenance and refurbishment have more or less a similar meaning. Work is done on a regular basis to keep the buildings and units in good condition and well painted. There is no rent increase and the work does not improve the comfort of the units. Renovation, sometimes called deep renovation, is done every 20–30 years. The comfort and rent both increase a consequence. There are many levels of renovation. However, the aim of most renovations nowadays is to provide better energy efficiency (assuming at least EPC = B) and indoor quality, as well as a decrease to energy cost. Renovations

usually deal with the building and not with the individual units. According to agreements with Aedes, the total for living expenses will remain the same (rents increase but energy costs decrease). Usually retrofitting means adding new heating and ventilation systems.

Depending upon the architectural quality of the building, the Amsterdam municipality chooses either for preservation or restoration street façades. Sometimes they demanded preservation and other times restoration as part of the renovation process. Particular pre-war buildings have had their street facades returned to a previous appearance.

Because this study deals with pre-war buildings, this means that the cases investigated have had their brickwork and wooden floors and roof renovated. For this reason, it is possible to change or merge units, implementing completely new floorplans. Renovation of this housing stock is only possible through an expensive renovation in uninhabited state. Because changes are made to the units and structure, renovation is done according to the criteria mentioned in the 'Bouwbesluit', the Dutch legislation on construction of buildings, which includes paragraphs on drastic renovation (ingrijpende renovatie). Local bylaws also need to be adhered to. Consequently, requirements for new buildings often guide renovation projects.

*1.6. Readers' Manual*

This article primarily describes the recent renovation of some eye-catching pre-war walk-up apartment buildings of housing associations in Amsterdam from the Gulden Feniks Award archives of the NRP [7]. Secondly, it describes the expensive and complex technique of a box-in-box renovation of these pre-war apartment buildings in uninhabited state [8–10]. Thirdly, it describes the renewed Housing Act of 2015, housing policy and consequences [11,12] and finally, recent developments and consequences for the Amsterdam municipality.

## 2. Beauty is Sustainable: Pre-War Walk-Up Apartment Buildings of Amsterdam

*2.1. The Golden Period of Renovation and Gentrification*

So what were the successes of renovating pre-war walk-up apartment buildings in what can be called the Golden Period from 1995 to 2015? As part of a wider gentrification policy, residential building renovations in diverse pre-war neighbourhoods of large Dutch cities began in the last decade of the twentieth century [13]. For the housing stock inside the ring road of Amsterdam there was a policy to improve and renovate old socially uniform workers' neighbourhoods for more socio-economic differentiation with residents of various lifestyles living side by side to strengthen the local economy and social coherence. Neighbourhoods were also transformed in Den Haag [14] and Rotterdam into attractive places to live. These cities wanted neighbourhoods to maintain their amenities, thus improving the future value of the residential district and buildings [15,16]. There were doubts about this approach [17]. Sometimes units were either sold off or let as free-market rental homes after the renovation of what had been affordable housing complexes. Note that attractive architectural icons were renovated. The Amsterdam municipality, the Amsterdam Federation of Housing Associations (AFWC), the Renter's Association of Amsterdam (HA) and other organisations between them came to an agreement and framework about this approach [18–20]. The historic façades at the street-side would be restored, but the garden-side, floor plans and in some cases also the private gardens inside the blocks could be completely changed by housing associations and tenant organisations. Gentrification was the result.

*2.2. Beginnings: Spaarndammerbuurt*

One of the great successes and iconic example project from the period 1995 to 2015 was the Spaarndammerbuurt in Amsterdam (Figure 1). The plan for renewal of this working-class neighbourhood, built in the period 1914–1920, took place from 2001 onwards. Most buildings there are of a in sturdy Brick Rationalism, four-storeys high with impressive pitched roofs. However, the most iconic of them was Het Schip by the architect M. de Klerk, a building from 1919 an icon of the Amsterdam School

of Expressionism. A pre-existing school that had been integrated in the block was transformed into a museum about how the working class lived during the inter-war period. After a deep renovation, some of the apartments were rented as free-market housing, some were sold and a part remained dedicated to social rentals. The municipality, housing associations and tenant organisations had two objectives: profiling for tourism and the promotion of a mix of lifestyles and income groups [21–25]. Participation commissions were organised by housing association tenants from different buildings and blocks.

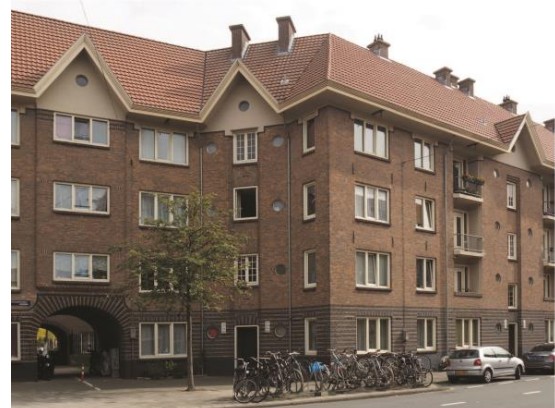
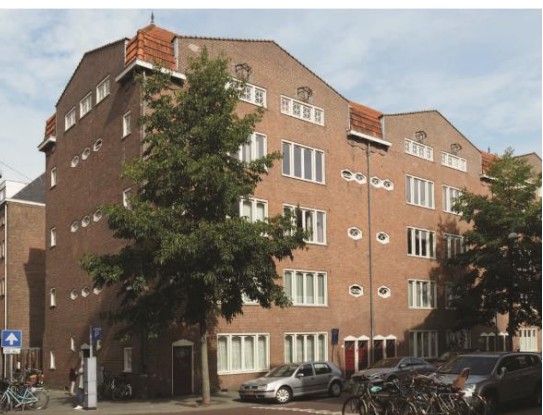
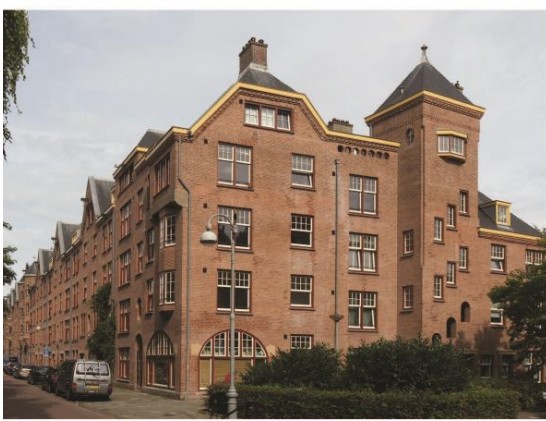
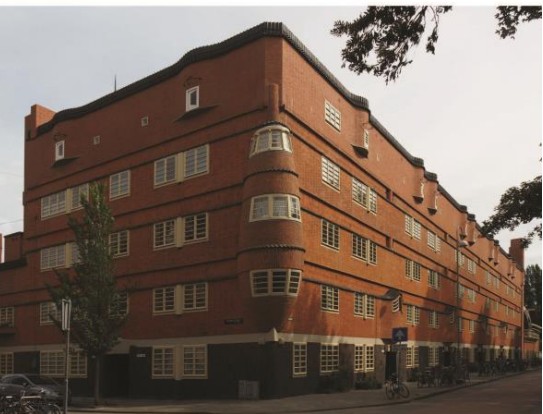

**Figure 1.** The walk-up buildings of Spaarndammer neighbourhood Amsterdam. **Top left:** Zaandammerplein by architect De Bazel (1918–1929), renovation by Archivolt architecten. **Top right** Spaarndammercarré by architects Gulden and Geldmaker (1916–1921), renovation by Hooyschuur architecten. **Bottom left:** Zaanhof Saenveste by architect Kuipers and Ingwersen (1915–1916), renovation by Hooyschuur architecten. **Bottom right:** Het Schip by architect Michel de Klerk (1914–1921), renovation by Archivolt architecten. Photos by Leo Oorschot.

About 71 per cent of the dwellings in the Spaarndammerbuurt were still affordable and rented by housing associations by 2014. The sale of a portion of the apartments financed the expensive deep renovation and conservation. The aim was not only to renovate the buildings, but also to change the unvaried population and create a neighbourhood with a higher average income with as result more amenities in the neighbourhood. The average price of an apartment in Amsterdam rose 17 per cent from €290,000 to €340,000 in the period between January 2017 and January 2018, according the CBS (Centraal Bureau voor de Statistiek) and the municipality. The change in the average sale price of dwellings of housing associations per district of Amsterdam in the period 2004–2017 was presented by the AFWC [12]. In 2004, the average price was €128,671 by 2017 it had increased to €273,847 [12]. The number of affordable housing association apartments shrunk between 1995 and 2017, especially in the pre-war neighbourhoods [12]. Apartments were enlarged or merged in most projects. On the one hand gentrification saved the quality of architecture and amenities of the neighbourhood and brought

a mix of people from more diverse social economical backgrounds. On the other hand, the tenant mobility and possibility to renovate other buildings stagnated.

Zaandammerplein Ensemble: three perimeter blocks and five buildings with 560 walk-up apartments were designed by the architect K.P.C. de Bazel and constructed around the Zaandammerplein square in the period 1918–1923 and 1926–1929. This urban ensemble is listed and considered as a monument of Dutch social history. The walk-up apartments were renovated in uninhabited state in the period 2005–10. The apartments are arranged in sets of four to eight, accessed through one shared stairwell. After the renovation, 100 apartments were rented out in the free market sector. Between 2006 and 2010 the average tax value or municipal valuation (WOZ) of these apartments increased by 40 per cent. The deep renovation strategy was to conserve the street façades as much as possible. The renovation applied the box-in-box isolation principle. Thermal and sound insulation and mechanical ventilation were provided. The floor plan was changed and apartments were combined. The storage space in the pitched roof was merged with the apartments, and collective heating was changed to individual heating with natural gas. Consequently the energy label of F/G was upgraded to C/B [26].

Spaarndammercarré Ensemble: Spaarndammercarré involves four corner buildings around a crossroad, built in the period 1916–1921 by the Socialistische Woningbouwvereniging Amsterdam-Zuid and designed by the famous housing architects Zeeger Gulden and Melle Geldmaker. The buildings were constructed in succession in three periods and were not designated as architectural heritage by the municipality. Some of the tenants returned to the first building's new affordable apartments without a rent increase. To finance the renovation, apartments in other buildings were either sold or let on the free market. The deep renovation strategy was similar to the buildings at Zaandammerplein: conservation of the street façade, box-in-box renovation to improve the indoor climate (soundproof, fire/smoke resistance and ventilation, achieving energy label A), new floor plans. Floor plans of 64 square meters were merged to 129 square meters. The garden façade was rebuilt and everything else within the blocks was changed, the courtyard redesigned and rebuilt. All individual gardens were eliminated and storage space was added as new volumes in the courtyard. A new collective garden was made on top of these new volumes. Balconies were replaced by a gallery, which was connected to lifts. Because the stairwell was removed as well, there were more usable square meters meaning more income for the housing association to finance the expensive renovation [27].

Zaanhof Ensemble Het Spoor: as a part of the Zaanhof ensemble two buildings with 93 apartments were redeveloped. As usual in Amsterdam inside the ring road, the renovation strategies was conservation of the street façade, complete new floor plans and improved indoor quality of the apartments with a low temperature heating, mechanical ventilation to achieve a B-rated energy label. Sound insulation was also improved. In this case, the garden façade was changed completely and a gallery and two lifts replaced the balconies and the stairwell was removed like at the Spaarndammercarré ensemble. After the renovation, the apartments were partly reserved as affordable units for returning tenants, a portion were rented out on the free market and the rest sold [28].

Zaanhof Ensemble Saenveste: another part of the Zaanhof ensemble is now called Saenveste. Originally, the 256 apartments were designed by Tjeerd Kuipers and Arnold Ingwersen between 1915–1916 and 1918–1919 and built by the Protestant housing association Patrimonium. The urban plan of the ensemble Zaanhof was designed by architect H.J.M. Walenkamp. He also designed some of the buildings and architect W. Greve designed the others. The renovation strategy here was again similar to other projects which lead to the existing label E/F being upgraded to B. In this case, the garden façade was insulated on the outside and no gallery and lifts were added. After the renovation, the apartments were again divided into affordable units for people who returned to their home, free market rental units and units for sale [29].

### 2.3. Koningsvrouwen Van Landlust Bos En Lommer Amsterdam

The building Koningsvrouwen van Landlust is designed according to CIAM principles by architect Gerrit Versteeg and built in 1937 in the neighbourhood Bos en Lommer, Amsterdam (Figure 2).

This project was an example of tenant participation during the process of renovation. The living cost remained the same but the quality of the apartments was improved. After a long period of preparations, the renovation started in 2007. Other aims were reduction of energy demand to less than 40 KWh/m²/year and conservation of the façades to show their original character. There were 134 walk-up apartments with an average of 46 net square meters before renovation, the rent was €246 and energy cost €186 a month, the sum total of which remained unchanged. The building has been transformed into 102 apartments and about 51 per cent of the tenants returned. The usual deep renovation strategy was applied to achieve energy label A: box-in-box renovation with a complete new floor plan and indoor quality, and mechanical ventilation. Sound insulation also got attention. Individual heating was removed and a new collective low temperature heating system applied: a geo-thermal heat pump and photovoltaic cells to provide electricity for the pump [30].

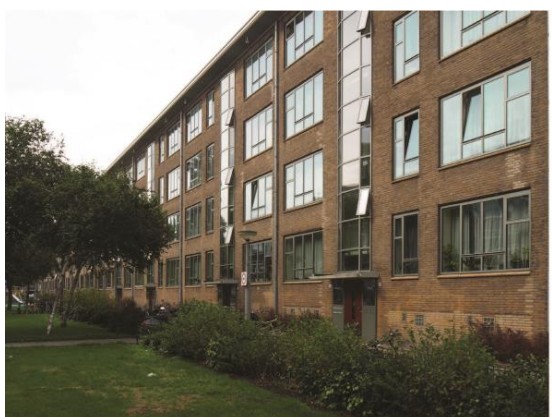
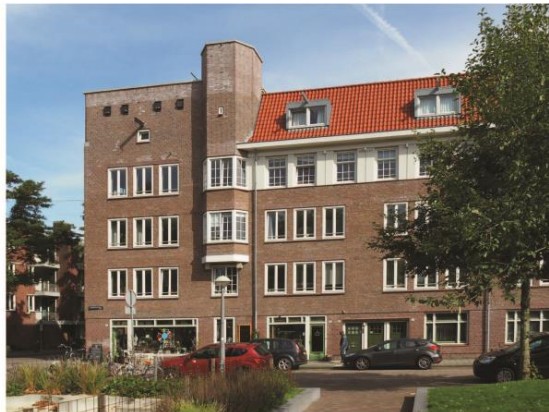
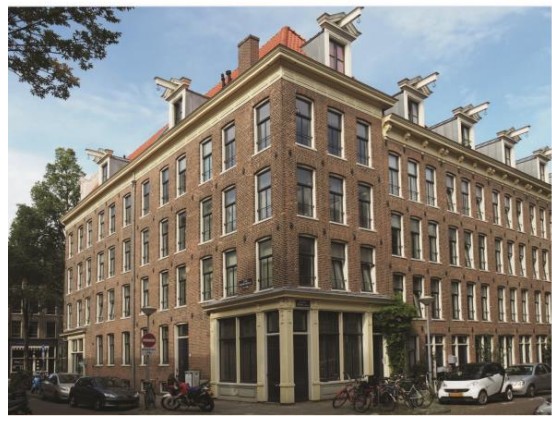
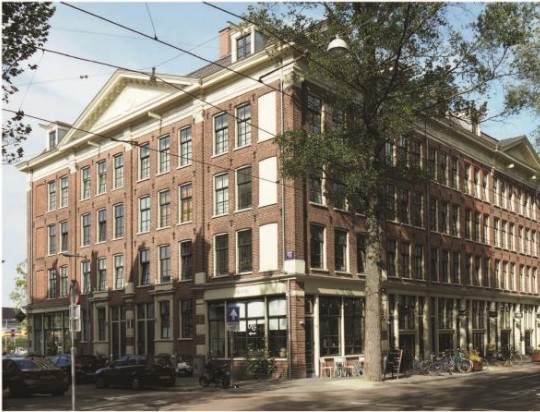

**Figure 2. Top left:** Koningsvrouwen van Landlust Bos en Lommer Amsterdam by architect Versteeg (1937), renovation by Archivolt architecten. **Top right:** Gorontalo Indische buurt Amsterdam, renovation by Van Schagen Architecten. **Bottom left:** Czaar Peterbuurt blok 46 Amsterdam, renovation by Van Ancum Hoogendoorn. **Bottom right:** Czaar Peterbuurt blok 42 Amsterdam, renovation by Hooyschuur Architecten. Photos by Leo Oorschot.

*2.4. Justus Van Effenblok Spangen Rotterdam*

One of the foremost apartment buildings from the inter-war period is the Justus van Effenblok in the working class neighbourhood Spangen in Rotterdam. It was designed by Michiel Brinkman and constructed in 1922. It is an urban block with a wide gallery on the garden side with lifts. It was once renovated in the eighties; however, the quality of the interventions according to the so-called 'Rotterdam method of urban renewal' ruined the architectural quality of the building. The 264 small apartments were merged into 164 larger apartments with a new floor plan. But the social problems did not disappear and a lot people left the building and the neighbourhood. With a tender (in Amsterdam

there were hardly any tenders in relation to renovation apartments), the architects Molenaar & Co in consortium with Hebly Teunissen Architects won the commission in 2005. There were two objectives for the deep renovation: a better indoor quality with a high level of energy efficiency and reinstating the quality and legibility of the original brickwork architecture, referring to the 'sustainable beauty of the architecture'. This meant the conservation of the facades and a box-in-box renovation to improve the indoor environmental quality. The 164 apartments were merged into 154 and the architects tried to bring back the quality of the original floor plans. For example, the kitchen, bathroom and toilet were relocated back to the façades on the gallery. Furthermore, the private gardens that had been introduced in the eighties were replaced by the original collective courtyard. Energy efficiency was based on reduction of the energy demand by an excellent insulation, low temperature hydronic heating and a collective heat pump with seasonal thermal energy storage STES, a solar collector on the roof for domestic hot water and photovoltaic panels. All affordable dwellings were changed to free market rental units and some of the apartments were sold to finance the expensive renovation. As in Amsterdam, the municipality of Rotterdam had the ambition to achieve more differentiation in social and economic background of the inhabitants of the neighbourhoods [31].

### 2.5. Gorontalo Indische Buurt Amsterdam

Here again, the aims of the municipality and the housing association were to achieve socio-economical differentiation of the inhabitants in the poor working class neighbourhoods couples with an energy efficient housing stock while respecting the architectural qualities of the buildings. The Gorontalo building on the Makassarplein was designed in the inter-war period by Jan Kuiler (Figure 2). The deep renovation strategy was as usual: conservation of the street façade, a box-in-box renovation (thermal and sound insulation, mechanical ventilation) with new floor plans and adaption of the façade on the garden side. Apartments were merged into larger units and the traditional shallow laundry balcony was extended to 1.5 m. In this case there was a heat pump connected to the mechanical ventilation system of the building, hydronic heating system with a boiler, solar collectors for domestic hot water and photovoltaic panels on the roof to provide electricity to the heat pumps. Label G/F of the apartments was improved to A/A+, however, as usual in this period, each apartment got an individual natural gas heating source. The apartments remained as part affordable social rentals and the rest were changed to free marked rental units [32].

### 2.6. Hoofdweg De Baarsjes Amsterdam

This building in the neighbourhood De Baarsjes was constructed in 1920 by a real estate developer and designed in the style of the Amsterdam School. However, when this building was renovated in 1973 and again in 1985, almost all original architectural details were removed. The architecture of the building was devastated and the façade became unattractive and poor. But with the new deep renovation of 2014, old details of the street façade could be restored. On the garden side, the shallow balconies were extended. Inside there was a box-in-box renovation with new floor plans and improved indoor quality (thermal and sound insulation, mechanical ventilation). Label G was converted to A. The storage space under the pitched roof was merged with the apartments. Of the 60 social rentals, only 24 remained as such and 34 apartments were sold. A part of the inhabitants returned to the renovated apartments [33].

### 2.7. Czaar Peterbuurt Amsterdam

Czaar Peterbuurt was once a poor and socially uniform working class neighbourhood (Figure 2). The walk-up apartments were built in the second part of the nineteenth century by private investment companies. At the end of the eighties, the first buildings were demolished and an urban renewal process began to re-house people from the neighbourhood. Some new buildings were added and an old factory transformed into housing. Housing associations Eigen Haard and Lieven de Key became the new owners of the buildings. In the nineties the neighbourhood suddenly changed.

New urban settlers found affordable units in the neighbourhood and new amenities appeared, such as a theatre (Theaterfabriek). In 2003 the borough adopted an approach based on differentiation of people with different incomes and dwellings. Especially the introduction of a new tramline in 2004 had an impact on the quality of the neighbourhood. Eigen Haard started several renovation projects: block N40, 41, 42, 45, 46 and 51. Furthermore, there was a change in attitude: the old buildings were not demolished and replaced but renovated instead. The Czaar Peterbuurt-development with 520 apartments and 50 shops is one of the large retrofit projects accomplished. Several old houses and other buildings were transformed into large apartments. The renovation started in 2009 and was completed in 2016. Some of the apartments were sold after the transformation. Urban block N46 (Czaar Peterstraat, Blankenstraat, Eerste Leeghwaterstraat, Tweede Leeghwaterstraat) was constructed in 1881 and involved 98 dwellings and 10 business spaces. In the period 2009–2011 the units were transformed to 63 dwellings and 11 business spaces. Apartments were merged. Label F and G were transformed with a box-in-box approach transformed to label B and A [34]. Urban block N42 has three parts (Conradstraat, Czaar Peterstraat, Lijndenstraat, Frans de Wollanststraat) and was constructed around 1850. The buildings were transformed into 76 dwellings and 13 business units. As usual the box-in-box approach was applied [35]. At the moment, most old dwellings in the Czaar Peterbuurt were renovated by housing associations.

## 2.8. Gentrification Was the Result

These eye catching projects were just illustrations of what happened in Amsterdam and Rotterdam. On a much smaller scale this process was applied in Den Haag (with many pre-war walk-up apartment buildings), and some in Rotterdam and Utrecht. Between 1995 and 2015 the change of the average selling price of existing houses, the ownership of these houses, and the number of houses that were sold was enormous, especially inside the ring road of Amsterdam [19]. Another cooperative agreement (Samenwerkingsafspraken) for the period 2015–2019 was reached between Amsterdam, AFWC and HA to allow housing associations to sell a maximum of 2000 dwellings per year and turn another 1000 to higher sector free-market rentals [20]. In the year 2015, housing associations sold 2337 homes [12]. Since then, the property sale inside the ring road has been sharply tempered. In the attractive Amsterdam neighbourhoods, charming pre-war walk-up apartment buildings were renovated to label B or even A by housing associations. Eye catching and sometimes listed walk-up apartment buildings entered for the NPR Gulden Feniks award were partially rented outside the affordable category of housing associations and a number were sold. The year 2015 was a turning point, after that year the amount of selling and renovations of inter-war walk-up apartment buildings diminished rapidly [12]. In their paper *Hoe zwak is een zwakke buurt?* about the Czaar Peterbuurt the authors Mariska van der Sluis and Pim Hogenboom (Eigen Haard) claimed that by this approach of renovation and the mixing of people an important function in the affordable regional housing market could disappear [36].

## 3. The Box-In-Box-Renovation-Approach

The integral box-in-box-renovation-approach of pre-war walk-up apartments with the characteristic brickwork facades and loadbearing walls within the Amsterdam ring road materialized into a success in the period 1995–2015, as Figures 1 and 2 shows. The NRP and housing association websites show deep renovations with conservation of the street façade, box-in-box renovation with new floorplans and service systems, and adaptation of the garden façades and courtyard, as Figure 3 shows. Not only was energy efficiency important, but several other issues were addressed, like fire safety, sound reduction between apartments and improved quality of the floor plans. This strategy is the result of an informal agreement and financial support from the City of Amsterdam, housing associations and tenant organisations. On a much smaller scale Rotterdam and Den Haag applied this strategy in this period as well. Furthermore, building regulations forced housing associations to apply box-in-box renovation because of fire/smoke resistance and soundproofing between apartments. The quality

of the street façade, degree of conservation or renovation is according to municipal guidelines [37]. As pointed out, stakeholders worked closely together according to agreements made.

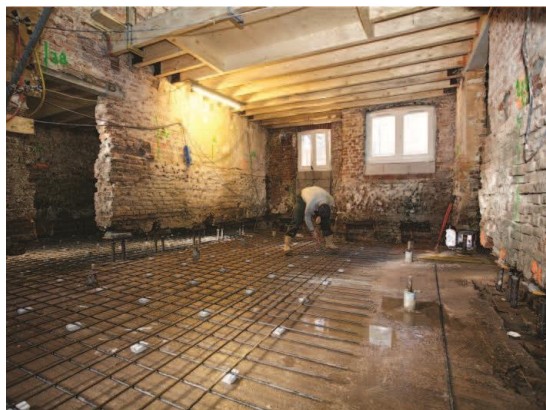
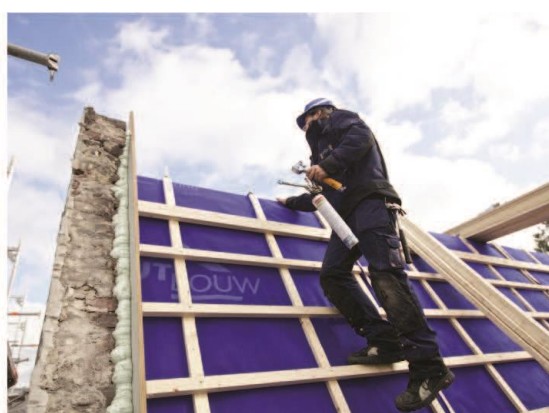
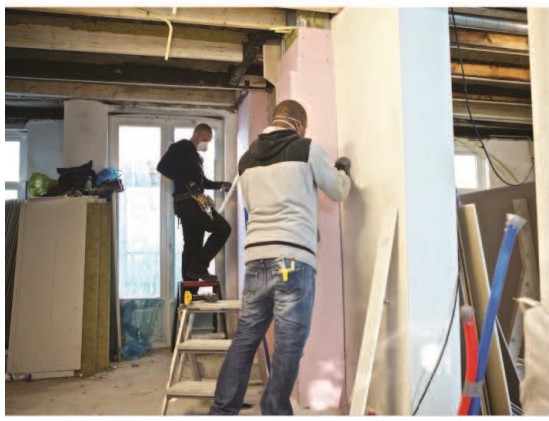
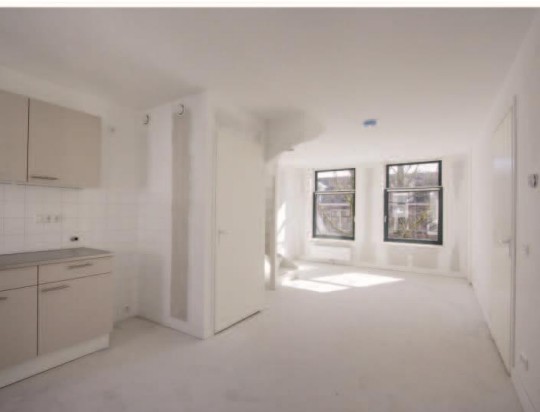

**Figure 3.** The expensive integral box-in-box-renovation of Czaar Peterbuurt block N46. **Top left:** Renovation of load bearing walls and foundations. **Top right:** Roof is replaced. **Bottom left:** Walls are covered. **Bottom right:** New apartment with a new floorplan. Photos by Mercuur Bouw.

The box-in-box-renovation usually comprised all the apartments around one stairwell in an uninhabited state. The attic, usually used for storage, was generally merged with another apartment. A part of the renovated and merged apartments were sold or became free sector rentals. If one stairwell was finished the next was started until all the tenement apartment blocks were completed. In some cases, private gardens were changed into storage blocks and a community roof garden was added. In some cases a gallery at the garden side with lifts replaced the stairwell. The inevitable choice for a box-in-box-renovation was based on a number of structural problems: [38–40].

- Functional: kitchen was separated from the living room
- Functional: bathroom was originally for the laundry and connected to kitchen and balcony
- Functional: lack of storage and service space in the apartments.
- Functional: the balcony on the garden side was to dry clothes outside and is too shallow as a functional outdoor space
- Structure: weak bearing walls and foundations
- Structure: outdated service systems, sewer pipes, ducts, and electric systems
- Materials: asbestos in several building components
- Materials: poor fire safety between apartments with wooden floors
- Skin: high energy demand because of poor thermal insulation and fit not being airtight

- Noise and moisture: after renovations in the seventies and eighties with new double-pane glass and airtightness of the façade, new problems appeared such as moisture problems and mould in apartments because of a natural ventilation system. Noise between apartments became more prominent due to the reduction of street noise after soundproofing the window units.

At this point it is interesting to note how the principle choice for the box-in-box approach has had a huge impact on the appearance of these housing blocks after renovation and, as a result, on the present quality of urban space in large parts of our major cities. This is different from the 'winter coat' approach that is implemented in most countries in our climate zone implies the addition of high-performance insulation materials on the outside and a new finish such as panelling or plaster. Typically this has a high impact on the architectural articulation due to loss of detail, proportion and material expression. Evaluation of early pilot projects in the Netherlands, for example in the neighbourhoods Moerwijk and Morgenstond or the listed neighbourhood *'t Fort* by the fin de siècle architect W.B. van Liefland in Den Haag, revealed that inhabitants were often unhappy with the results of deep renovations. In many cases, the early weathering of the new surface materials appeared, often resulting in a shabby and poor image.

The urban and architectural qualities of social housing in the Netherlands are relatively high. Several pre- and post-war social housing schemes are listed as national heritage and many others are designated as municipal monuments by local governments. But even the unlisted residential blocks are often appreciated for their architectural merit by professionals as well as the public at large, albeit perhaps to various degrees. In that sense the outside appearance of these structures form part of the collective memory or our city dwellers. Recent research indicates that present residents do not necessarily support an aesthetic makeover of their building if there are no other benefits involved [10]. Although developed to resolve the abovementioned technical problems in an integrated way, the box-in-box approach also opened up opportunities to retain the architectural features of these buildings to a higher degree.

Inside the Amsterdam ring road, housing associations renovated the pre-war walk-up apartments with a financial depreciation period of 30 or 40 years. With box-in-box-renovation the integral interventions are collectively addressed: thermal and sound insulation, fire safety, floor plan improvements and infrastructure replacements. The method makes use of dry construction and new pipes and channels are concealed in the new walls. The sustainability goal implies low temperature heating so that apartments change from Energy Performance Certificate F-G to B-A grading. For central heating and domestic water a Natural Gas Heating Water Boiler is usually installed. The ventilation system is changed from natural to mechanical. Fire and smoke resistance, sound reduction and ventilation of existing apartments are fitted according to building regulations for new housing. A box-in-box approach means that within existing structures of old bearing walls and wooden floors, a new box is created that reduces noise, fire, smoke, and energy-use demands. The floating floors, suspended ceilings and all walls of the apartment are insulated. Because repairing the loadbearing walls and foundations and addressing moisture problems are necessary in Amsterdam, the wooden ground floor construction is usually replaced by insulated concrete. The advantage of this box-in-box-renovation is that floor plans can be changed and small apartments merged at the same time. According to the NRP archives, RVO database, and guidelines of Eigen Haard [38–40], there are different priorities among the different interventions.

1. Integral indoor climate approach: improve the skin of the building and reduce energy demand, aiming for label B or higher, applying low temperature heating (LTH) and mechanical ventilation, if possible with demand control ventilation (DCV) [41,42].
2. Apply an advanced hydronic or non-hydronic heat system (various heat sources mentioned). Because of the limited capacity of all these systems, these options will most likely be in use by 2050 (Figure 4):

   - District heat network, or in the near future, a regional open heat network (70 degrees Celsius) with Industrial Waste Energy (EfW) or Geothermal Energy for about 20 or 30 years.
   - Local heat network (50 degrees Celsius) with various local heat sources, such as Industrial Waste Energy (EfW) or Geothermal Energy, Urban Waste Heat Sewage, or other local heat sources.

- All-electric heating with a collective or individual heat pump with a 50 degree Celsius output, with air (ASHP), water (WAHP), or the earth (GSHP) as the heat source. The central heating (CV) is LTH and domestic tap water is heated with a booster with a heat pump; a small electric heater is still necessary in wintertime. New and expensive types of heat pumps have an output of 70 degrees Celsius and a LTH is not necessary. This is an inefficient way of providing heat for the heating system because only during the winter peak does the heat pump use cold air as a heat source. Like non-hydronic heat systems, there is the option of all-electric heating with, among others, electric heating of the house and a solar collector to provide domestic hot water.
- All-electric heating with an individual heat pump with air as the heat source, and a non-hydronic system that directly heats the air inside the smaller unit.

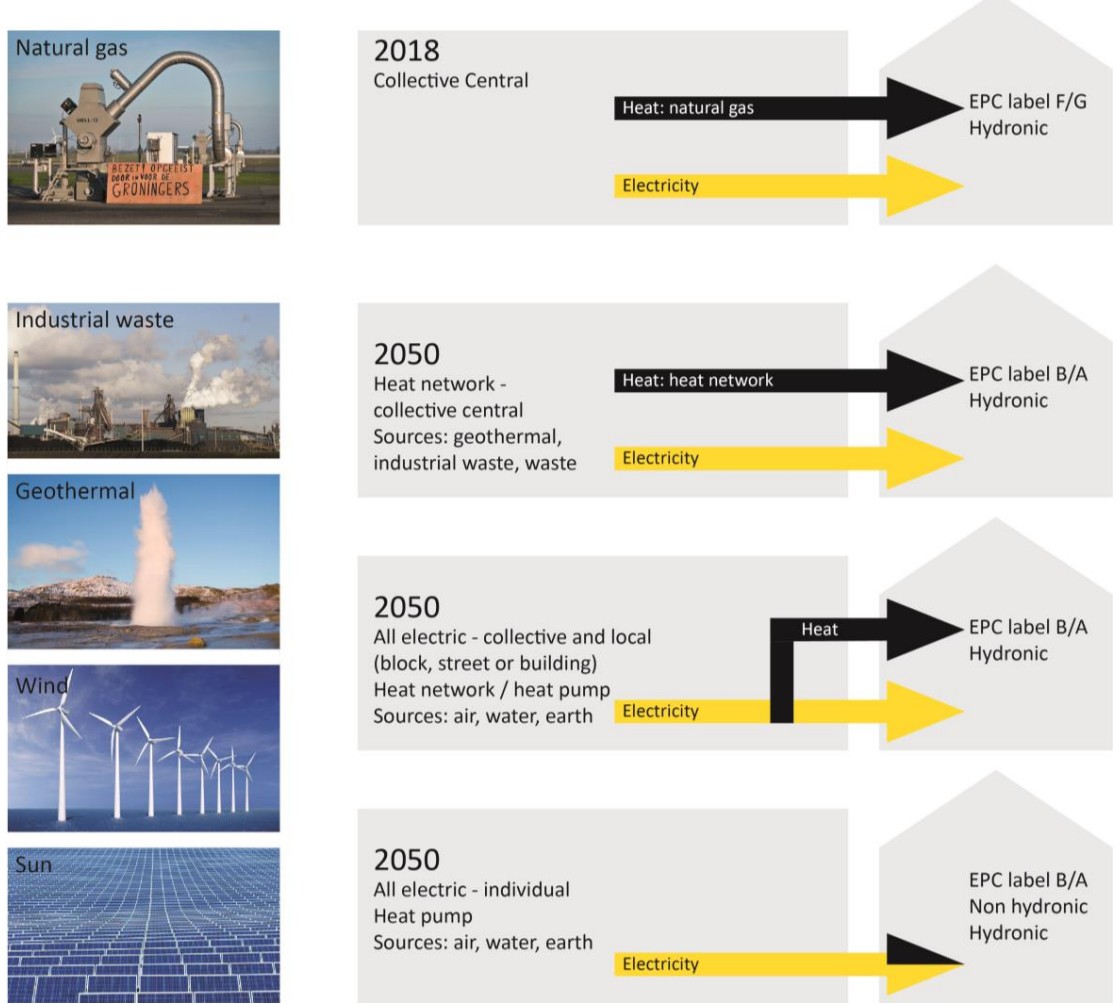

**Figure 4.** Various options providing energy to the existing housing stock by 2050. Graphic Leo Oorschot.

The bio- or synthetic option is difficult because there is not enough natural gas to supply all the dwellings. Furthermore, the gas infrastructure must be replaced anyway because of the age of much of the infrastructure. Laying out a heat network is difficult because the infrastructure will be very expensive and to reduce the cost, and at least 70% of a neighbourhood of about 50–300 dwellings must be connected to the network. This option is complex and expensive because of the spread of individual private ownership in pre-war neighbourhoods and the limited available space in streets and courtyards of the pre-war urban ensembles and finding the space for a heat station and space for the infrastructure demanded is very difficult. In fact, the all-electric option is the only feasible option left for these

neighbourhoods. However, at this moment the electric grid does not have the capacity to provide the required amount of electricity. Furthermore, there are noise problems with heat pumps, especially if they are on balconies located on courtyards. Heat pumps with air as heat source produce between 40 and 50 dB(A). In forthcoming regulations, 35 dB(A) is assumed and expensive noise reduction measure are therefore needed. Photovoltaic panels could deliver electricity to the heat pump during the day. Additionally, there are also rather inefficient options with pipes and ducts as heat source, for example sewer system, mechanical ventilation or showers with heat recovery. During the winter peak these system could not provide the energy required for the dwellings. For these reasons, reducing the energy demand and changing behaviour are still important. The conclusion is that all-electric exploitation is technically possible if one is prepared to invest in additional interventions to reach the insulation level needed for LTH. The described cases show that an EPC label A or higher is possible with an Energy-Index between 0.71 and 1.05. But, so far there are no renovations of a walk-up apartment building from this period that apply the all-electric principle.

With pre-war walk-up apartment buildings within the Amsterdam ring road in the period 1995 to 2015 there was the box-in-box-approach with a focus on energy efficiency and the reduction of energy demand by insulating buildings and the use of natural gas. As pointed out, the focus after the Paris-Climate-Change-Conference 2015 is to change from energy efficiency to carbon neutrality in construction and operation of buildings. The Netherlands aims to be carbon neutrality by 2050. However, after 2015 housing associations in Amsterdam only renovated some small projects aimed at EPC B [43,44]. Similar developments can be seen in Den Haag and Rotterdam. But is it necessary to invest more in the reduction of energy demand by renovating the skin of the building to the level of label B or A. According to the current building regulations for new homes their goal is an EI =< 0,4 (EPC A+++).

Residential energy consumption has two components; the energy demand related to heating or cooling depends on the level of insulation of the skin of the building – the EPC – and the energy demand related to the behaviour and size of the household. A household with children uses much more hot domestic water than someone who lives alone [10]. If a walk-up apartment has label B or A, then the energy cost/demand of a household with children is probably more than the cost/demand of heating. Recent studies confirm that the behaviour of tenants is important. There is a performance gap in the prediction of energy consumption before and after the deep renovation because of changing use of the dwelling and behaviour by tenants [45].

A deep renovation aiming at label B (1,21 < EI <= 1,4) with 51 to 90 kWh/m$^2$/year energy use or label A (0,8 < EI < 1,2) with less than 50 kWh/m$^2$year in combination with renewable energy sources will be sufficient to reach the level of carbon neutrality. Because of the shape of the walk-up apartment buildings, there is not much space on the roof to have photovoltaic panels or solar collectors; therefore, apartment blocks always need an external renewable energy source for heating or cooling.

## 4. Tipping Point 2015

The change of the Housing Act of 1 July 2015 had a profound influence on the renovation of Dutch housing stock. It led to a lack of investment capacity for housing associations; a new points-based system was introduced to calculate the rent, constructions cost increased with the stagnation of new projects as result; tax-value became a parameter and led to an increase of the rent and higher taxation of housing associations (through a high tax-value, urban areas became very expensive and suburbs with a low tax-value less expansive). The flow rate of tenants is low and there is a lack of affordable units. Seize and income of the household became a parameter in calculating the rent.

Figure 5 shows the number of dwellings of housing associations sold in 2017 in each borough of Amsterdam. Figure 6 shows the change of the housing stock after ownership in Amsterdam in the period 1983-2017. Figures 7 and 8 shows the difference and number of sales of dwellings sold by housing association in Amsterdam in the period 1998-2017. Figure 9 shows the change in the average sales price of dwellings of housing association in Amsterdam. These figures were published by AFWC in 2018 [12].

| Number of sold corporation rentals per borough to buildings type | | | | |
|---|---|---|---|---|
| Borough | Unknown houses | Row houses | Multy-storey | Total houses |
| Centrum | 0 | 0 | 61 | 61 |
| West | 3 | 0 | 230 | 233 |
| New West | 1 | 45 | 75 | 121 |
| Zuid | 2 | 2 | 107 | 111 |
| Oost | 3 | 9 | 145 | 157 |
| Noord | 0 | 59 | 146 | 205 |
| Zuidoost | 0 | 16 | 78 | 94 |
| Total | 9 | 131 | 842 | 982 |

| Sold to the tenants per borough in pre cent and number | | |
|---|---|---|
| Borough | Per cent houses | Number houses |
| Centrum | 11% | 7 |
| West | 10% | 24 |
| New West | 5% | 6 |
| Zuid | 10% | 11 |
| Oost | 9% | 14 |
| Noord | 11% | 23 |
| Zuidoost | 9% | 8 |
| Total | 9% | 93 |

| Number of sold corporation rentals per borough for social and free rentals to individuals and companies | | | | | | |
|---|---|---|---|---|---|---|
| Borough | To individuals | | To companies | | Total social rentals | Total units |
| | Social rentals | Free rentals | Social rentals | Free rentals | | |
| Centrum | 52 | 9 | 0 | 0 | 52 | 61 |
| West | 187 | 46 | 9 | 0 | 196 | 242 |
| New West | 86 | 35 | 0 | 0 | 86 | 121 |
| Zuid | 78 | 33 | 0 | 1 | 78 | 112 |
| Oost | 105 | 52 | 0 | 4 | 105 | 161 |
| Noord | 164 | 41 | 0 | 0 | 164 | 205 |
| Zuidoost | 75 | 19 | 1 | 220 | 76 | 315 |
| Total | 747 | 235 | 10 | 225 | 757 | 1.217 |

**Figure 5.** The number of dwellings of housing associations sold in 2017 in each borough of Amsterdam. Districts Centrum, West, Zuid, and partly East are located inside the ring road (blue).

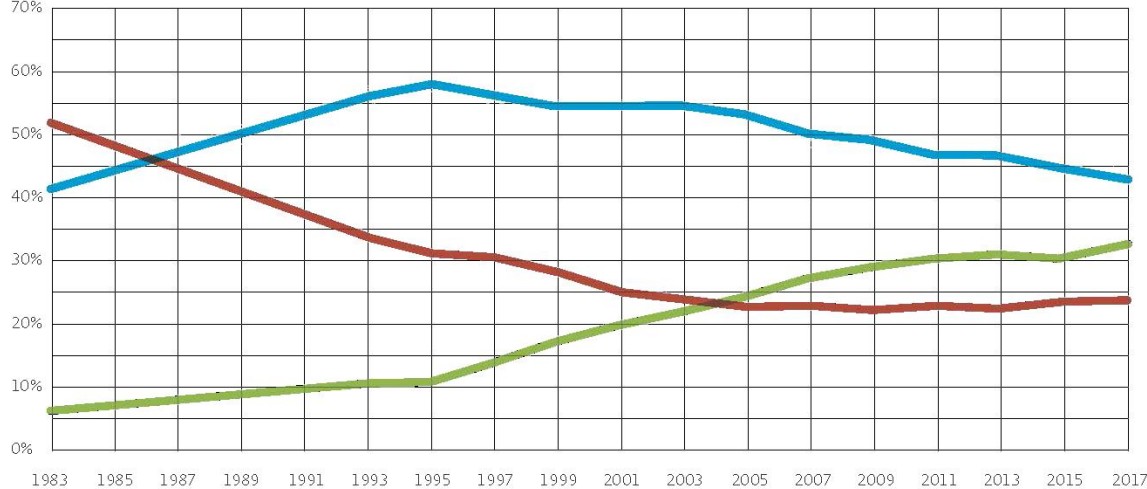

**Figure 6.** This Figure shows the change of the housing stock of Amsterdam after ownership in per cent between 1983 and 2017. Blue: social rentals housing associations. Red: free market rentals. Green: owner-occupied dwellings. Within the framework of urban renewal of the 1970s and 1980s, old houses were expropriated, demolished, and replaced by new apartment buildings. In 1995, the Housing Act changed, and housing associations were transformed into a kind of real estate developers. Just a portion of the dwellings were social rentals (usually 30 per cent) and another portion was sold.

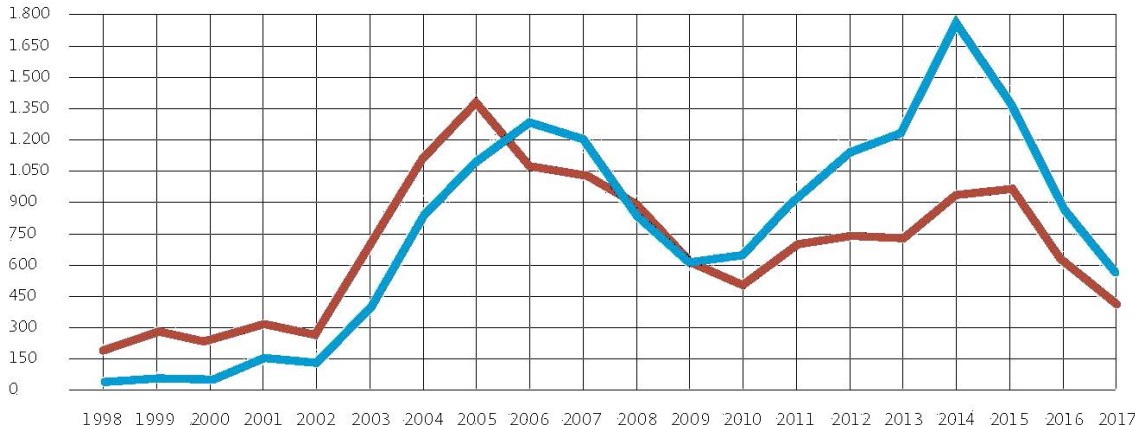

**Figure 7.** This figure shows the difference and number of sales of dwellings sold by housing associations inside (blue: the boroughs Centrum, Zuid, Oost and West) and outside (red: Nieuw West, Noord and Zuidoost) the ring road of Amsterdam between 1998 and 2017.

|  | Centrum | West | New West | Zuid | Oost | Noord | Zuidoost | Total |
|---|---|---|---|---|---|---|---|---|
| 1998 | 32 | 29 | 8 | 0 | 0 | 4 | 206 | 279 |
| 1999 | 51 | 4 | 96 | 0 | 22 | 15 | 188 | 376 |
| 2000 | 14 | 9 | 67 | 8 | 38 | 102 | 85 | 323 |
| 2001 | 2 | 66 | 78 | 52 | 52 | 164 | 88 | 502 |
| 2002 | 6 | 35 | 44 | 55 | 52 | 105 | 137 | 434 |
| 2003 | 47 | 140 | 65 | 70 | 141 | 322 | 283 | 1.068 |
| 2004 | 114 | 320 | 231 | 180 | 211 | 448 | 398 | 1.902 |
| 2005 | 119 | 405 | 417 | 171 | 375 | 558 | 357 | 2.402 |
| 2006 | 68 | 596 | 257 | 146 | 437 | 490 | 302 | 2.296 |
| 2007 | 94 | 531 | 205 | 203 | 342 | 470 | 326 | 2.171 |
| 2008 | 55 | 362 | 141 | 148 | 264 | 428 | 306 | 1.704 |
| 2009 | 53 | 232 | 145 | 116 | 208 | 277 | 183 | 1.214 |
| 2010 | 84 | 256 | 142 | 99 | 199 | 189 | 175 | 1.144 |
| 2011 | 77 | 382 | 132 | 229 | 214 | 364 | 186 | 1.584 |
| 2012 | 156 | 431 | 202 | 254 | 266 | 324 | 195 | 1.828 |
| 2013 | 124 | 394 | 215 | 330 | 349 | 342 | 161 | 1.915 |
| 2014 | 192 | 594 | 258 | 570 | 412 | 436 | 220 | 2.682 |
| 2015 | 176 | 473 | 272 | 369 | 355 | 453 | 239 | 2.337 |
| 2016 | 139 | 251 | 174 | 257 | 254 | 268 | 181 | 1.524 |
| 2017 | 61 | 233 | 121 | 111 | 157 | 205 | 94 | 982 |
| Total | 1.664 | 5.743 | 3.270 | 3.368 | 4.348 | 5.964 | 4.310 | 28.667 |

**Figure 8.** The number of dwellings sold in the period 1998–2017 by housing associations after a deep renovation to individuals of each borough of Amsterdam. Most units were sold in the periods 2005–2007 and 2014–2015. Sales to investment companies are not included. Borough Centrum, West, Zuid and partly East are inside the ring road (blue).

| | 2004 | 2005 | 2006 | 2007 | 2008 | 2009 | 2010 | 2011 | 2012 | 2013 | 2014 | 2015 | 2016 | 2017 |
|---|---|---|---|---|---|---|---|---|---|---|---|---|---|---|
| West | 153.611 | 161.016 | 164.843 | 174.551 | 195.160 | 195.205 | 210.180 | 190.666 | 174.773 | 176.578 | 183.543 | 214.029 | 264.970 | 285.173 |
| New West | 138.867 | 140.074 | 155.135 | 150.787 | 158.779 | 165.098 | 160.564 | 160.829 | 161.569 | 154.040 | 156.206 | 174.812 | 207.660 | 244.274 |
| Zuid | 182.408 | 182.560 | 200.872 | 213.178 | 230.504 | 211.975 | 197.074 | 201.674 | 186.443 | 189.278 | 183.260 | 226.069 | 272.342 | 347.825 |
| Oost | 161.787 | 156.250 | 162.263 | 183.201 | 193.205 | 180.190 | 180.481 | 184.742 | 171.117 | 185.128 | 186.073 | 214.723 | 276.858 | 318.764 |
| Noord | 136.556 | 137.391 | 147.736 | 150.166 | 158.089 | 157.333 | 162.881 | 163.700 | 149.739 | 150.144 | 149.277 | 159.918 | 208.317 | 232.684 |
| Zuid-oost | 128.671 | 128.453 | 127.313 | 132.498 | 134.940 | 131.925 | 132.810 | 133.734 | 124.777 | 129.107 | 135.578 | 132.285 | 151.489 | 178.826 |
| Total | 149.354 | 148.464 | 157.733 | 167.875 | 176.132 | 172.841 | 178.061 | 176.852 | 167.533 | 169.750 | 171.905 | 193.079 | 239.159 | 273.847 |

**Figure 9.** This Figure shows the change in the average sales price of dwellings of housing associations per borough of Amsterdam in the period 2004–2017. In 2004, the average price was €128,671. This increased to €273,847 by 2017. Borough Centrum, West, Zuid and partly East are inside the ring road (blue).

### 4.1. Lack of Investment Capacity

Before the last economic crisis and the announcement of the new Housing Act, housing associations preferred a demolition and replacement approach for many four-storey apartment buildings, as one of their objectives was to stimulate differentiation in household income in uniform working class neighbourhoods. Between 1995 and 2015, housing associations usually built 30 per cent of their units for lower incomes and 70 per cent for other income groups. Housing associations financed the production and reconstruction of affordable houses with more expensive housing. The updated Housing Act of 2015, however, stimulates housing associations to focus their activities only on lower income households, so instead of demolition they now try to maintain the affordable housing stock and emphasize renovation strategies. New units for people with a middle income are not built anymore. Nonetheless, the current neo-liberal government claims that while the focus must be on lower income groups, they still treat housing associations like companies. For example, housing associations and landlords with more than 50 housing units are now charged a fee (verhuurdersheffing) based on the average tax value or municipal valuation (WOZ) by the government and tenants who live in affordable dwellings but earn a too high income face considerable rent increases. The average tax value (WOZ) is related to the real estate value in the Netherlands [46–49]. As early as 2008 housing associations were treated like companies and taxed as such (venootschapsbelasting). Additionally, housing associations are charged with a new European Anti-Tax Avoidance Directive (ATAD) after 2019. For these reasons, the investment capacity of housing associations has diminished radically over the last years.

### 4.2. The New Point System (WWS) and Tax Value or Municipal Valuation (WOZ)

When tenants move, they also face higher rentals according to the point system (WWS) introduced in 2013 in which one of the parameters used in calculating the rent is the tax value of the real estate object. The real estate value is a function of the quality of the neighbourhood and its location in the municipality. For instance, the rent is higher for an apartment inside the ring road of Amsterdam than a similar apartment in the countryside. Moreover, the high pressure on the housing market in larger cities makes it difficult to move tenants out for renovation or demolition purposes. As pointed out, deep renovation of the pre-war housing stock is only possible in uninhabited state. Furthermore, in areas with high pressure on the housing market, like Amsterdam, there is a gap between the rent calculated with the point system (WWS). Since 2019 for persons or household with an income above a certain income limit currently set at €38,035 (80% of all social dwellings) and €42,436 (10% of all social dwellings) a year and the maximum rent of €720.42 (under the income limit) to be eligible for funding by the state. Housing associations call this a rebate. If tenants have an income above a certain limit this rabat disappears. But even then, it is still a lower rent than a free market rental in Amsterdam inside

the ring road. The flow rate of tenants was already very low in attractive residential areas near historic centres. No one wants to leave his or her home for one with a higher rental as determined by the new point system (WWS) [46–49].

### 4.3. Construction Cost in the Future

At the same time, requirements for sustainability have been further tightened, with consequences in the cost of construction and of infrastructure. Retrofitting apartments has become expensive, and will become even more expensive in the future. Municipalities are now mandated to have a transition vision on heat (Transitievisie Warmte) and an energy plan for each neighbourhood (Wijkenergieplan) in their city by 2021. But what will happen with the affordable pre-war walk-up apartment buildings of Amsterdam and Den Haag? Which strategy will be chosen and what are the consequences for the affordable housing stock? Currently the strategy is a renovation that aims at achieving energy label B or A utilising a gas boiler as heating source, but with preparations made to disconnect the natural gas later on. Yet connecting to a future heat network of one apartment or one building individually is not an option because this required good collective organisation. Furthermore, not every housing association currently has the investing capability to renovate their housing stock to energy label B. Deep renovation of pre-war apartment buildings is costly because of the repair of the loadbearing walls and the foundations of the blocks. All-electric solutions are now still too expensive for a regular deep renovation in the social rental sector. Another problem is that buildings of the inter-war period under the care of housing associations are spread over different locations and neighbourhoods with many different real estate owners. Reaching agreement on a heat network will be a challenge. In contrast, post-war apartment buildings in housing association ownership are generally concentrated in ensembles in just a few neighbourhoods. For that reason a heat network is a more likely option here [50]. Furthermore, if the aim is for an all-electric or a heat network energy supply of walk-up apartment buildings, one could draw the conclusion that someone will have to pay for the investments. It is likely that the tenant will be paying for the investments of the new energy infrastructure as part of the energy cost. Rotterdam and Den Haag have already decided to sell their share in the energy company Eneco, the owner of the local heat network. The multinational Shell will probably be the next owner. In the Amsterdam region, the owner of the heat network Nuon is already part of the Swedish Vattenfal company. Both are private companies selling energy.

### 4.4. Flow Rate of Tenants and Lack of Affordable Units

The current stagnation of tenant mobility and the need for affordable dwellings is an important reason to develop renovation strategies that are feasible while allowing inhabitants to stay in their apartments. This is especially for post-war walk-up apartments. However, with pre-war walk-up apartments this is almost impossible, as pointed out. If households have to be moved, each receives a legally established moving fee of €5,910. Furthermore, under Dutch legislation, 70% of the tenants in an apartment complex must agree with the proposed renovation plans and the change in rent. Nobody agrees to move from the attractive centre of Amsterdam to a faraway suburb. As Figure 6 shows, in Amsterdam, with a relatively large housing stock (ca. 45%) belonging to housing associations and a great shortage on the housing market, plus a policy of mixing housing categories, housing associations have sold their apartments or rented them out expensively. However, in Den Haag with much less social housing (ca. 30%), the sale of social housing has been undesirable, with as consequence that tenement apartment blocks have been renovated at a much slower rate in Den Haag. But the biggest problem is that there is hardly any replacement housing available for people and few affordable dwellings have been added to the housing stock, as Figure 6 shows. Because of all this uncertainty, the flow rate of tenants remains low. In crisis municipalities, housing associations and municipal governments are trying to stimulate people to move by implementing flow plans like 'Van Groot naar Beter' in Amsterdam, and similar in Den Haag. With a roundtable discussion, 'Samenwerkingstafel', in 2017, the government and stakeholders can to an agreement with stakeholders about the flow of

tenants [51]. Since the renewed Housing Act of 2015 took effect, new houses are only being assigned to tenants with appropriate low incomes. Consequently, tenants with low incomes are gradually being concentrated in neighbourhoods with tenement apartment buildings.

### 4.5. Manifest Passend Wonen

On 7 June 2018, 23 housing associations of Amsterdam proposed and published the *Manifest Passend Wonen* [52,53]. They introduced a new parameter for calculating the increase of rent. Not only will the household income be checked, but the number of people in the household will be an included as parameter in calculating rentals in the future. This will be monitored annually. In other words, the appropriate assignment of new units to tenants, their incomes and composition of the household among others, will become new parameters in Amsterdam. The reason is that too many tenants with a middle or high income are occupying the dwellings in the attractive pre-war neighbourhoods of the city, even though they are paying the maximum rent according to the point system. The housing associations of Amsterdam calculated that about 70,000 households (30 per cent) in Amsterdam have a too high an income for the units they live in and 10,000 households without children have apartments with three or more bedrooms. In Den Haag 35,000 household have a too high an income for the units they live in [49]. But all efforts have been in vain. The increase of rent for an apartment from housing associations near the historic centre of Amsterdam or Den Haag remains less than a free marked rental. In the future, besides the income of tenants, is also the composition of the household (the number of people in relation to seize of the unit) a parameter [54,55].

### 4.6. Selling, Buy-To-Let, Airbnb

Furthermore, according to the municipality of Amsterdam 57% of all dwellings in the city were affordable houses (<€71,068) in 2018 (belonging to both housing associations and private companies) and a lack of middle category rentals (€71,068–€97,100) at just 6% of the housing stock. The policy for the coming years up to 2025 is to equalise these figures in order to arrive at 40 per cent each [51]. Housing associations and real estate companies have been organised into a platform to realize this ambition (Platform Amsterdam Middenhuur, PAM) [50]. The policy for the near future in Amsterdam is that housing associations will only sell units when more than 35 per cent of housing units in a neighbourhood are located to lower income tenants [51]. It is likely for several reasons that in the future many pre-war walk-up apartment buildings in former working class neighbourhoods will be sold to tenants and real estate companies, as Figure 5 shows. As a consequence of the renewed Housing Act of 2015, the point system (WWS) and the challenge of a $CO_2$-neutral housing stock, among others, tenants have to pay more rent in attractive neighbourhoods nearby the historic centre, than in a suburb. Furthermore by 2018 the real estate and tax value (WOZ) increased by 17 per cent in Amsterdam, with consequences for to the rents of apartments. With buy-to-let and Airbnb, real estate prices are out of control in Amsterdam [46–49,56].

## 5. Conclusions

In this study the object of research is the pre-war walk-up apartment buildings inside the ring road of Amsterdam and the phenomenon of stagnation of deep renovation of this housing stock. Therefore, the main research questions were:

What was the successful approach to renovation in the period 1995–2015? It seems that the approach in Amsterdam between 1995 and 2015 was only possible because of pressure on the housing market and the huge appreciation of real estate in its beautiful historic residential neighbourhoods, part of which was sold or became free market rentals. This explains why the pre-war apartment blocks in Amsterdam were renovated and not in Den Haag, the other city with a lot of pre-war walk-up apartments. In Den Haag with 30 per cent social housing, already decided a few years ago not to sell dwellings. In Amsterdam, as Figure 6 shows, housing associations will continue to sell dwellings until they only have 40 per cent social housing stock left. New agreements between housing

associations, tenant organisations and the municipality have led to a decrease in sales of affordable housing. However, Figure 5 shows 842 multi-storey apartment sales in 2017, most of which were located in the inter-war districts West (230), Zuid (107), and Oost (145) [12].

What is the influence of the European EPBD on the renovation stagnation in Amsterdam? Requirements about the EPC were tightened. However, because of the complex box-in-box-renovations and the requirements for indoor quality (new or merged floorplans), an integral renovation approach was necessary anyway in order to extend the life span of the historic buildings and provide safe (fire and smoke) and healthy ventilated units.

How is the renovation stagnation influenced by national legislation (such as the change of the Housing Act of 2015)? The Renewed Housing Act of 2015 and several other changes as mentioned in paragraph 4, above, have had an enormous impact on the housing market. This is one of the reasons why housing associations have partially sold off their pre-war real estate. Apartment buildings have changed from collective owners (housing associations) to private owners. To renovate these apartment buildings, a collective approach that accommodates continuous inhabitation, as pointed out in paragraph 3, with the box-in-box-renovation approach is necessary. Incorporating a heat network is only possible if there is this kind of collective. Furthermore, due to the renewed Housing Act of 2015 and the consequential shrinking housing market for low and middle incomes, stagnation of tenant mobility occurred, which directly led to the stagnation of renovations of this housing stock. In addition, living expenses (rent and energy costs) make it impossible to assign the pre-war walk-up apartment units appropriately to people in relation to their income, with segregation as a consequence. Housing associations sell their housing stock inside the ring road of Amsterdam and build new affordable houses in distant suburbs. This is the only way housing associations can provide affordable housing for their target group.

What is the cause of the increase in living expenses, such as rent and energy costs? As pointed out, the urban changes are complex and energy efficiency is surely one of the parameters. As mentioned in paragraph 4, the newly introduced point system (WWS) which includes the tax value or municipal valuation (WOZ) as a parameter are factors that led to an increase of the rent in attractive urban areas (usually inside the ring road of the city) and decrease of rent in unattractive parts of the municipality (usually outside the ring road). Figure 7 shows that almost two times more units were sold inside (blue line) than outside (red line) the ring road.

What are the consequences of these developments for Amsterdam and other municipalities? The consequence for Amsterdam is the segregation of people and a stagnation of the renovation of pre-war walk-up apartment buildings, regardless of whether there will be all-electric or a heat network by 2050. The feasibility of the heat network is diminished option; housing associations guarantee a collective approach but with every other unit having another owner, implementing a heat network—which requires collectivity—will be problematic.

Finally, as recently pointed out by two Amsterdam based housing associations CEOs (Marien de Langen of Stadgenoot and Leon Bobbe of De Key), the tax value (WOZ) on property affects tenants in attractive urban regions for two reasons. Firstly, the fee (verhuurdersheffing) that housing associations have to pay to the government is based on the tax value of their property. For that reason housing association have less investment capacity for the renovation or construction of new units in urban regions. Secondly, the rent tenants have to pay to housing associations is partially based on the tax value of the unit. The average tax value in the Netherlands is €750.00 per dwelling per year, while in Amsterdam it is €1200.00 a year [57]. These property taxes significantly influence the rental price in attractive city neighbourhoods.

**Author Contributions:** L.O. conceptualized and wrote this manuscript, W.D.J supervised the manuscript.

**Funding:** This work is part of the research program Research through Design, project number 14569, which is (partly) financed by the Netherlands Organization for Scientific Research (NWO) and Taskforce for Applied Research SIA.

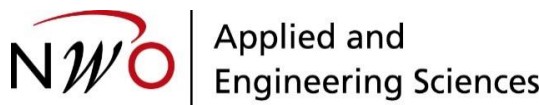

**Acknowledgments:** The current research was carried out through the design project of the Delft University of Technology, Faculty of Architecture and the Built Environment, and University of Applied Sciences Utrecht between May 2016 and May 2018, and was about user preference-tested design solutions for cultural values and energy-efficient walk-up apartment building renovations of inter- and postwar buildings in Dutch city regions. The research team members were Leo Oorschot, Sabira El Messlaki, Thaleia Konstantinou, and Clarine van Oel. Research was done with housing associations, institutes, and experienced renovation architects. From the Delft University of Technology, Vincent Gruis, Thijs Asselbergs, Lidwine Spoormans and Wessel de Jonge were involved.

**Conflicts of Interest:** The authors declares no conflict of interest. The founding sponsors had no role in the design of the study; in the collection, analyses, or interpretation of data; in the writing of the manuscript; or in the decision to publish the results.

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
