# Peer review of "Progress and Stagnation of Renovation, Energy Efficiency, and Gentrification of Pre-War Walk-Up Apartment Buildings in Amsterdam Since 1995"

_sustainability, doi:10.3390/su11092590_

Reviewer 1 Report

The current version of the paper has improved a lot.

I believe that it may be published

Author Response

Dear reviewer,

Thank for the remarks on the text. We made some changes to the paper to clarify my arguments, especially the introduction and the conclusion.

About the method and sources we use. The method that was applied in this research is a survey of annual statistics provided by housing associations of Amsterdam (AFWC) about their housing stock and tenants, and annual project data files provided by the institute Platform voor Transformatie en Renovatie (NRP) about deep renovation projects. Furthermore, documents published by the government, housing associations, municipalities, consultancies and institutes were used as sources to interpret the data. These documents clarify why and in which way these buildings were renovated, and why changes took place.

NRP (Platform voor Transformatie en Renovatie) is a Dutch institute and organisation that deals with knowledge and education about the renovation process of buildings and the transformation of the built environment. It is supported by real estate companies, the Delft University of Technology, architects, consultancies, housing associations and construction companies. Each year, the NRP organises the Gulden Feniks Award and collects project data files about the renovation of this housing stock from the participants competing for the award [see the reference 7].

Another important source are the annual statistics about the housing stock and tenants, which are provided by the Amsterdams Federatie Woningcorporaties (AFWC). This is the federation of the nine Amsterdam based housing associations that owned 186,429 dwellings altogether as of 2018 [see the reference 12]. This is about 42% of the total amount of dwellings in Amsterdam. These statistics present key information about migration, segregation and diversification, and social housing in Amsterdam. The data is used by the municipality, tenant organisation of Amsterdam (HA) and housing associations to reach an agreement about the local housing policy (Woonvisie) in accordance with national legislation.

In addition, we paid attention to the renewed Housing Act of 2015, which had an impact on the housing market in the Netherlands. Primary literature was studied to understand the changes in the practice of renovation of the housing stock. For example, protocols about the renovation strategies of housing associations such as Eigen Haard [see the references 38, 39, 40].

This study brings different fields of knowledge (statistics, documents, protocols, project data files) together and draws some conclusions about the stagnation of the renovation of pre-war apartment buildings of housing associations.

Kind regards.

Reviewer 2 Report

Abstract

The abstract has described the issues or problems well, however it is not informative enough for people to understand what the aim of the project is and methodology for the research. The abstract should be re-written to capture very succinctly the what, why, how and so what? The abstract is also very disjointed and could be improved in English grammar.

Line 15: change ‘prewar’ to ‘pre-war’ and also change in all subsequent occurrence of this

Line change: just make sure you mean ‘revaluation’ not ‘re-evaluation’

Lines 13 to 18: ‘Urban renewal, that was traditionally based on demolition and replacement with new buildings changed in improvement of old buildings’ consider rephrasing to ‘Urban renewal, that was traditionally based on demolition and replacement with new buildings has changed to improvement of old buildings through refurbishment’

Introduction

A lot of readers may not be familiar with the term ‘walk-up apartments’ perhaps you should explain early in the intrioduction

Line 51. ‘BENG’-norm, what is this? You should define it the first time you use an acronym

Line 52: The definition of NZEB should be better articulated>

Line 61: ‘milliard’ kilogram … is this a misspell?

Line 69: ‘However, is the increase living expense of tenants only caused by this challenge?’ because you are starting a totally new section, you should consider rephrasing to ‘Therefore, the main research question is ‘is the increase in living

Line 69-71 ‘For example tenants of municipalities with a lot attractive historic buildings like Amsterdam shows that there are many more reasons why the living expenses such as rent and energy cost are increasing’ perhaps you should provide evidence of this?

Line 79: What is NPR? Please define this the first time you refer to it.

Research methods

‘The method that was applied in this research is a survey of refurbished prewar walk-up 78

 apartment buildings’ further explanation need to be provided on what the term ‘survey’ means in the context of this research, is it in terms of ‘building survey’ or ‘occupant survey’ etc.

Line 86: ‘NRP is the organisation that deals with knowledge about transformation process of buildings and the built environment’ this is the first time you defined what ‘NRP’ is even though you have referred to them several times, you should define all abbreviations and organisations and their role within the project life cycle or as stakeholders.

Some of the definitions of ‘retrofit’, ‘refurbishment’, ‘renovation’, ‘deep renovation’ could have been backed up by some evidence from the literature, unless this is arguably your interpretation.

Line 107: spelling mistake ‘assocaitions’ should be ‘association’

Line 108: ‘(1.1.2017)’ is this a reference? Is it a mis-spell? If it is a reference, it is not in the reference list. 

Line 145: ‘Amsterdam School Expressionism….’ Is this ‘Amsterdam School of Expressionism’.

Case study description

The case study developments have been described very well, however perhaps before the description you could also explain and summarise what is it you are trying to find in terms of similarity and differences between the various developments, this should guide and focus the discussion of the case study developments.

Line 157: ‘The average price of an apartment in Amsterdam rose from €256.000 to €359.000 in the period 2013 to 2016’ Reference is required here.

Line 319: ‘AFWC and HA’ what does this mean? Define in full.

Line 482: ‘ Figuur 5’ spelling mistake

The quality of all tables and figures can be improved for clarity.

Conclusions

The conclusion should be improved to show clearly how the conclusion has been derived from all the primary and secondary data collected in the research.  

Author Response

(The authors gave the same response as above.)

Reviewer 3 Report

The work presents an analysis of the reasons for the increase of cost and energy in prewar apartment buildings in Amsterdam.

From my point of view, the work is too much descriptive, rather than holding a scientific character and it can not be published in the present form. The information is presented through narrative text, instead of providing tables of figure that summarise and synthesise it, enclosing a more scientific perspective.

In my opinion, the findings have little impact for the scientific community.

Research questions are not well stated/addressed and conclusions do not support them.

Author Response

Dear reviewer,

Thank for the remarks on the text. We made some changes to the paper to clarify my arguments, especially the introduction and the conclusion.

About the method and sources we use. The method that was applied in this research is a survey of annual statistics provided by housing associations of Amsterdam (AFWC) about their housing stock and tenants, and annual project data files provided by the institute Platform voor Transformatie en Renovatie (NRP) about deep renovation projects. Furthermore, documents published by the government, housing associations, municipalities, consultancies and institutes were used as sources to interpret the data. These documents clarify why and in which way these buildings were renovated, and why changes took place.

NRP (Platform voor Transformatie en Renovatie) is a Dutch institute and organisation that deals with knowledge and education about the renovation process of buildings and the transformation of the built environment. It is supported by real estate companies, the Delft University of Technology, architects, consultancies, housing associations and construction companies. Each year, the NRP organises the Gulden Feniks Award and collects project data files about the renovation of this housing stock from the participants competing for the award [see the reference 7].

Another important source are the annual statistics about the housing stock and tenants, which are provided by the Amsterdams Federatie Woningcorporaties (AFWC). This is the federation of the nine Amsterdam based housing associations that owned 186,429 dwellings altogether as of 2018 [see the reference 12]. This is about 42% of the total amount of dwellings in Amsterdam. These statistics present key information about migration, segregation and diversification, and social housing in Amsterdam. The data is used by the municipality, tenant organisation of Amsterdam (HA) and housing associations to reach an agreement about the local housing policy (Woonvisie) in accordance with national legislation.

In addition, we paid attention to the renewed Housing Act of 2015, which had an impact on the housing market in the Netherlands. Primary literature was studied to understand the changes in the practice of renovation of the housing stock. For example, protocols about the renovation strategies of housing associations such as Eigen Haard [see the references 38, 39, 40].

This study brings different fields of knowledge (statistics, documents, protocols, project data files) together and draws some conclusions about the stagnation of the renovation of pre-war apartment buildings of housing associations.

Kind regards,

Round  2

Reviewer 2 Report

The manuscript has been improved significantly, see below some editorial changes:

Line 44: change ‘kWh per m2’ to kWh/m2 also change in subsequent places.  

Line 47: Eedes, if this is a short form, write in full. 

Line 187: Spell check ‘trnasformed’  

Line 364: figure, capitalise the first letter 

Line 370: …..diminished rapidly [12]. Stagnation follows.... room for clarification here..  

Line 386 – Line 388: Figure 3 caption has been deleted from manuscrift, insert new caption.

Author Response

Dear reviewer, thanks for you remarks on the article. I must admit, it is not an easy topic Energy transition and gentrification of the old social housing stock, among many other aspects, has a profound impact on our cities today. The last changes of the text are in red. Kind regards.

Reviewer 3 Report

I mantain my previous decision about the publication of the paper. In my opinion the research is too much descriptive, rather than providing a scientific basis.

Author Response

Dear reviewer, I must admit, it is not an easy topic Energy transition and gentrification of the old social housing stock, among many other aspects, has a profound impact on our cities today. The last changes of the text are in red. Kind regards.

This manuscript is a resubmission of an earlier submission. The following is a list of the peer review reports and author responses from that submission.